

# Rate dependent neural responses of interaural-time-difference cues in fine-structure and envelope

Hongmei Hu[1,2], Stephan D. Ewert[2], Birger Kollmeier[2] and Deborah Vickers[1]

[1] SOUND Lab, Cambridge Hearing Group, Department of Clinical Neuroscience, Cambridge University, Cambridge, United Kingdom
[2] Department of Medical Physics and Acoustics, Carl von Ossietzky University of Oldenburg, Oldenburg, Germany

Corresponding author
Hongmei Hu, hh594@cam.ac.uk,
hongmei.hu@uni-oldenburg.de

## ABSTRACT

Advancements in cochlear implants (CIs) have led to a significant increase in bilateral CI users, especially among children. Yet, most bilateral CI users do not fully achieve the intended binaural benefit due to potential limitations in signal processing and/or surgical implant positioning. One crucial auditory cue that normal hearing (NH) listeners can benefit from is the interaural time difference (ITD), *i.e.*, the time difference between the arrival of a sound at two ears. The ITD sensitivity is thought to be heavily relying on the effective utilization of temporal fine structure (very rapid oscillations in sound). Unfortunately, most current CIs do not transmit such true fine structure. Nevertheless, bilateral CI users have demonstrated sensitivity to ITD cues delivered through envelope or interaural pulse time differences, *i.e.*, the time gap between the pulses delivered to the two implants. However, their ITD sensitivity is significantly poorer compared to NH individuals, and it further degrades at higher CI stimulation rates, especially when the rate exceeds 300 pulse per second. The overall purpose of this research thread is to improve spatial hearing abilities in bilateral CI users. This study aims to develop electroencephalography (EEG) paradigms that can be used with clinical settings to assess and optimize the delivery of ITD cues, which are crucial for spatial hearing in everyday life. The research objective of this article was to determine the effect of CI stimulation pulse rate on the ITD sensitivity, and to characterize the rate-dependent degradation in ITD perception using EEG measures. To develop protocols for bilateral CI studies, EEG responses were obtained from NH listeners using sinusoidal-amplitude-modulated (SAM) tones and filtered clicks with changes in either fine structure ITD ($ITD_{FS}$) or envelope ITD ($ITD_{ENV}$). Multiple EEG responses were analyzed, which included the subcortical auditory steady-state responses (ASSRs) and cortical auditory evoked potentials (CAEPs) elicited by stimuli onset, offset, and changes. Results indicated that acoustic change complex (ACC) responses elicited by $ITD_{ENV}$ changes were significantly smaller or absent compared to those elicited by $ITD_{FS}$ changes. The ACC morphologies evoked by $ITD_{FS}$ changes were similar to onset and offset CAEPs, although the peak latencies were longest for ACC responses and shortest for offset CAEPs. The high-frequency stimuli clearly elicited subcortical ASSRs, but smaller than those evoked by lower carrier frequency SAM tones. The 40-Hz ASSRs decreased with increasing carrier frequencies. Filtered clicks elicited larger ASSRs compared to high-frequency SAM
tones, with the order being 40 > 160 > 80 > 320 Hz ASSR for both stimulus types. Wavelet analysis revealed a clear interaction between detectable transient CAEPs and 40-Hz ASSRs in the time-frequency domain for SAM tones with a low carrier frequency.

# INTRODUCTION

Cochlear implants (CIs) have successfully restored the sensation of hearing to individuals with severe to profound deafness, and there are now over 1 million implant recipients worldwide (*Zeng, 2022*). Over the past 10–15 years, there have been improvements in advanced implant features, accompanied by a substantial increase in the number of patients, predominantly children, receiving bilateral CIs. These changes have led to significant improvements in CI outcomes, such as better speech understanding in noisy environments, and improved sound localization. However, there are still substantial differences in performance between bilateral CI users and normal hearing (NH) listeners in various binaural tasks (*e.g.*, *Laback et al., 2004*; *Nopp, Schleich & D'Haese, 2004*; *Grantham et al., 2007*; *Seeber & Fastl, 2008*; *Litovsky, Parkinson & Arcaroli, 2009*; *Litovsky et al., 2012*; *Hu et al., 2018*). Potential limitations could be imposed by the signal processing used to deliver the electrical stimulation and/or from the way the implants were surgically positioned. One crucial auditory cue that NH listeners can benefit from is the interaural time difference (ITD), representing the time difference between the arrival of a sound at two ears. The sensitivity for using these cues is believed to be heavily dependent on the effective utilization of temporal fine structure (rapid oscillations in the sound carrier), and partly on the utilization of temporal envelope (slow fluctuations superimposed on carrier) (*e.g.*, *Smith, Delgutte & Oxenham, 2002*). Unfortunately, most current CIs do not accurately transmit this true fine structure. Nevertheless, bilateral CI users have demonstrated sensitivity to both ITD (*Ihlefeld, Kan & Litovsky, 2014*; *Francart et al., 2015*; *Kan, Jones & Litovsky, 2015*; *Egger, Majdak & Laback, 2016*) and interaural level difference (ILD) (*Best, Laback & Majdak, 2011*; *Stakhovskaya & Goupell, 2017*). For example, bilateral CI users showed pulse ITD sensitivity, when the ITD cues were delivered through interaural pulse time differences, representing the time gap between the pulses delivered to the two implants. However, their ITD detection thresholds are typically 5–10 times higher compare to NH listeners, and the ITD sensitivity further degrades at higher CI stimulation rates, particularly when the rate exceeds 300 pulses per second (pps) (*van Hoesel & Tyler, 2003*; *Majdak, Laback & Baumgartner, 2006*; *Laback, Majdak & Baumgartner, 2007*; *van Hoesel, 2007*; *Poon et al., 2009*; *Ihlefeld et al., 2015*; *Kan & Litovsky, 2015*; *Laback, Egger & Majdak, 2015*). It should be noted that in the CI literature, the ITD sensitivity sometimes alternately uses the terms pulse ITD and envelope ITD (the ITD cues delivered through the envelope). The elevated ITD threshold and the rate—dependent ITD sensitivity of bilateral CI users are very similar to the findings in the envelope ITD sensitivity of NH listeners

(*Henning, 1974*; *Nuetzel & Hafter, 1981*; *Hafter & Dye, 1983*; *Bernstein & Trahiotis, 2002*; *Goupell, Laback & Majdak, 2009*; *Majdak & Laback, 2009*; *Stecker & Brown, 2010*; *Bernstein & Trahiotis, 2013*; *Stecker, 2014*; *Monaghan, Bleeck & McAlpine, 2015*). Various animal (*Griffin et al., 2005*; *Smith & Delgutte, 2008*; *Hancock et al., 2010*; *Chung, Hancock & Delgutte, 2016*; *Vollmer, 2018*; *Rosskothen-Kuhl et al., 2021*) and computer models (*e.g.*, *Colburn et al., 2009*; *Chung, Delgutte & Colburn, 2015*; *Hu, Klug & Dietz, 2022*; *Müller et al., 2022*; *Hu et al., 2023a*, *2023b*) have been used to understand the mechanisms underlying this rate-dependent degradation in bilateral CI users. However, the quantification of neural ITD sensitivity—reflecting neural responses to ITD cues—in CI listeners, particularly in relation to pulse rate, has not been systematically explored using non-invasive electroencephalogram (EEG) measures. This is, in part, due to the challenges associated with removing CI stimulation artifacts (*e.g.*, *Hofmann & Wouters, 2010*; *Hu & Dietz, 2015*; *Hu, Kollmeier & Dietz, 2015*; *Hu & Ewert, 2021*). Portions of this study were previously published as a preprint in *Hu et al. (2023c)*.

Numerous binaural EEG and magnetoencephalography (MEG) paradigms have been proposed to assess binaural hearing by measuring neural responses to different binaural cues, at different levels of the auditory pathways (*e.g.*, *Dobie & Berlin, 1979*; *Ross et al., 2007*; *Ross, Tremblay & Picton, 2007*; *Ross, 2008*; *Grose & Mamo, 2012*; *Hu & Dietz, 2015*; *McAlpine et al., 2016*; *Ozmeral, Eddins & Eddins, 2016*; *Papesh, Folmer & Gallun, 2017*; *Shinn-Cunningham et al., 2017*; *Eddins & Eddins, 2018*; *Vercammen et al., 2018*; *Gnanateja & Maruthy, 2019*; *Koerner et al., 2020*; *So & Smith, 2021*; *Ungan, Yagcioglu & Ayik, 2020*). In this study, we hypothesize that the spatial hearing is related to the neural encoding of ITD cues, such as the fine structure ITD ($ITD_{FS}$) and the envelope ITD ($ITD_{ENV}$), at different stages of the auditory pathway. Therefore, the $ITD_{FS}$ and $ITD_{ENV}$ sensitivities were obtained from the NH participants, which serves as a benchmark for the future bilateral CI experiments. To assess the binaural hearing, the MEG paradigm reported by *Ross et al. (2007)*, *Ross, Tremblay & Picton (2007)* was adapted to EEG measurements in this study. Three different EEG experiments were conducted in the same participants, with different types of ITD cues, respectively.

Experiment 1 aimed to validate the test setup and reproduce the $ITD_{FS}$ sensitivity reported in the literature. Previous studies have shown that the ability to detect a specific ITD value decreases as the carrier frequency increases above a certain threshold below 1,500 Hz (*Zwislocki & Feldman, 1956*; *e.g.*, *Ross et al., 2007*; *Ross, Tremblay & Picton, 2007*; *Grose & Mamo, 2010*; *Hopkins & Moore, 2010*; *Brughera, Dunai & Hartmann, 2013*; *Füllgrabe et al., 2017*; *Papesh, Folmer & Gallun, 2017*; *Füllgrabe & Moore, 2018*; *Ross, 2018*; see review, *Moore, 2021*; *Klug & Dietz, 2022*). Furthermore, the upper limit of ITD processing has been suggested as an indicator of binaural hearing deficit, because it is known to decreased in middle-aged and older adults (*Ross et al., 2007*). It has been utilized to diagnosis binaural temporal processing abilities, including age-related changes *via* MEG, EEG and behavioral measures (*e.g.*, *Ross et al., 2007*; *Ross, Tremblay & Picton, 2007*; *Grose & Mamo, 2010*; *Füllgrabe et al., 2017*; *Füllgrabe & Moore, 2017*; *Papesh, Folmer & Gallun, 2017*; *Ross, 2018*; see review, *Moore, 2021*). In this study, the EEG responses of $ITD_{FS}$ sensitivity were measured using sinusoidal amplitude modulated (SAM) tones at

four carrier frequencies (400, 800, 1,200, and 1,600 Hz) with modulation frequency of 40 Hz.

Experiment 2 aimed to investigate the causes of an absent acoustic change complex (ACC) response reported by *Ross (2018)* when evoked by a change in envelope ITD. *Ross (2018)* employed the same MEG paradigm as *Ross, Tremblay & Picton (2007)* to examine the responses to $ITD_{ENV}$ changes using 40 Hz SAM tones at carrier frequencies of 250, 500, 1,000, 2,000, and 4,000 Hz. Surprisingly, only two out of 14 participants exhibited significant responses to $ITD_{ENV}$ changes when the carrier frequency was set at 4,000 Hz. Although Ross and his colleagues did not explicitly use the term of ACC (*Ostroff, Martin & Boothroyd, 1998*; *Martin & Boothroyd, 1999*) in their studies, we are puzzled by the fact that the majority of their participants did not exhibit ACC responses to $ITD_{ENV}$ changes, despite their ability to perceive these changes. Considering the substantial interest in ACC for its role in objectively assessing monaural auditory discrimination capabilities (*Mathew et al., 2017*; *Mathew et al., 2018*; *Han & Dimitrijevic, 2020*; *McGuire et al., 2021*; *Undurraga et al., 2021*; *Calcus, Undurraga & Vickers, 2022*), and given that ACC responses are regarded as robust and indicative of a perceived change in an ongoing stimulus (*Martin, Tremblay & Korczak, 2008*), we find it challenging to comprehend the absence of ACC responses in the majority of participants in *Ross*'s *(2018)* study. Ross suggested that the $ITD_{ENV}$ processing at the subcortical level relies on stimulus phase locking, and individuals who showed responses at higher frequencies (2,000 and 4,000 Hz) might depend on cues other than the $ITD_{ENV}$. Given that responses evoked by $ITD_{ENV}$ for SAM tones at low carrier frequencies (<2,000 Hz) are primarily delivered through the fine structure (phase locking), experiment 2 utilized SAM tones with a high carrier frequency of 4,000 Hz. Building upon the findings of *Ross (2018)* and considering our preliminary pilot data, we recognize that the perceptually perceived changes in $ITD_{ENV}$ might not be sufficiently salient to evoke robust and easily detectable ACC responses. Consequently, experiment 2 employed multiple modulation rates (40, 80, 160, and 320 Hz) to further assess the impact of modulation rates on auditory steady-state responses (ASSRs) for potential future applications, even if ACC responses may not be easily detectable. This decision is motivated by previous physiological studies indicating age-related decreases in the ability to follow the highest modulation rates, as indicated, for example, by the envelope following responses (*Purcell et al., 2004*).

Experiment 3 aimed to simulate CI performance in NH listeners by testing the $ITD_{ENV}$ sensitivity of bandpass-filtered pulse trains as described in *Hu et al. (2017)*, *Hu, Klug & Dietz (2022)*, with a center frequency of 4,000 Hz and bandwidth of 2,000 Hz. EEG responses to multiple pulse rates (40, 80, 160, and 320 pps) were measured to assess whether filtered clicks will evoke larger responses than the corresponding SAM tones used in experiment 2. In various published studies, bandpass-filtered clicks have been employed to simulate CI performance in NH listeners (*e.g.*, *Laback et al., 2004*; *Carlyon, Long & Deeks, 2008*; *Hu et al., 2017*; *Hu, Klug & Dietz, 2022*). It was expected that these bandpass-filtered clicks would evoke larger responses than the SAM tones (*Hu, Klug & Dietz, 2022*).

The Multi-information extraction technique, which involves applying comprehensive approaches to simultaneously extract various types of information from a given dataset, was employed to obtain cortical and subcortical responses at different rates from the same participants. These responses include ASSRs, and the cortical auditory evoked potentials (CAEPs) elicited by stimuli onset, offset, and ITD changes in the ongoing stimulus. Considering the absence of detectable ACCs in the 12 out of 14 participants in *Ross (2018)* for the 4,000 Hz SAM, we were aware of potential challenges in measuring the ACC in experiments 2 and 3, especially given the use of a 'one-for-all' EEG data analysis pipeline without fine tuning for individual participants and conditions. This concern is magnified in the context of a clinical paradigm where typically only 2–4 electrodes are available, and the time constraints are present. However, with this multi-information extraction technique, even if the ACC responses evoked by the $ITD_{ENV}$ changes are not detectable, we still expected to obtain useful information from other types of responses, *e.g.*, onset, offset CAEPs, and the ASSRs at multiple modulation rates. Our assumptions are follows: 1) all test conditions in experiment 1–3 will elicit both onset and offset CAEPs. 2) Clear ACC responses will be evident in experiment 1 for conditions with $ITD_{FS}$ changes in carrier frequencies up to 1,200 Hz. 3) in experiment 2, ACC responses evoked by $ITD_{ENV}$ changes are expected to be either absent or smaller than those observed in experiment 1. 4) If detectable ACC responses are present in both experiment 2 and 3, it is anticipated that the ACC responses to the $ITD_{ENV}$ changes in bandpass-filtered clicks will be larger than those elicited by the corresponding high carrier frequency SAM tones used in experiment 2. 5) ASSRs will be detectable in all three experiments, at least for conditions with modulation rates below 320 Hz. Given the same modulation rate, ASSRs will be larger for conditions with carrier frequencies up to 1,600 Hz in experiment 1 compared to those with a carrier frequency of 4,000 Hz in experiments 2 and 3. 6) time-frequency analysis, such as wavelet analysis, offers additional information to the time or frequency domain alone. The unique advantage of this study lies in being the first EEG investigation to simultaneously obtain multiple cortical and subcortical responses within the same EEG recordings, covering various rates while assessing both $ITD_{FS}$ and $ITD_{ENV}$ sensitivity in the same participants. By ensuring a high degree of similarity in experimental setup and participant conditions across three experiments, the data generated from this study will provide invaluable insights, deepening our understanding of potential divergences in the mechanics governing $ITD_{FS}$ and $ITD_{ENV}$ sensitivity along the auditory path way.

## MATERIALS AND METHODS

### Participants

Eight NH participants (S1–S8: five males and three females aged 21–35 years old, with a mean age of 26.4 years) participated in the EEG experiments. None of the participants had a history of neurological, psychiatric, or otological disorders. All had audiometric thresholds of 20-dB hearing level or better at octave frequencies between 125 Hz and 8 kHz. Participants provided voluntary written informed consent and were compensated with hourly pay for their participation, with the approval of the Ethics Committee of the University of Oldenburg (Drs.EK/2019/075).

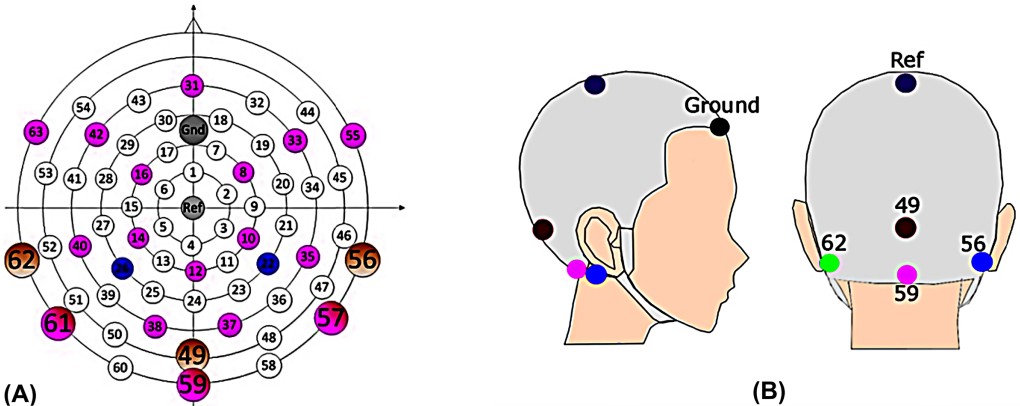

**Figure 1 EEG cap.** (A) The scalp channel locations and labels of the EEG cap (*Hu & Dietz, 2015*). The position on the central anterior-posterior line corresponds to a 10% electrode system, with electrode labels such as Fpz (31), Ref (Cz), and Iz (49), *etc.* Channels 49, 56, 59, and 62 were the channels of primary interest. (B) Electrode location for clinic setup. Mastoid (left: green, 62; right: blue, 56); Middle line (inion: 49; magenta, 59); reference (Cz, ref); ground (forehead).

## Apparatus

The Alternative Forced Choice (AFC) software (*Ewert, 2013*), a modular framework designed for running psychoacoustic experiments and computational perception models, and freely available, was used to control the experiments and present the stimuli to the participants. During the EEG experiment, a Fireface UCX sound card was connected to Tucker Davis (Alachua, FL, USA) HB7 headphone driver and presented to participants through ER-2 insert earphones (Etymotic Research, Elk Grove, IL, USA). The stimuli were calibrated to 75 dB SPL. Participants sat in a recliner and watched silent, subtitled movies in an electrically shielded soundproof booth while instructed to minimize movements. The EEG data were recorded differentially from Ag/AgCl electrodes *via* CURRY7 (Neuroscan) and a recording computer. Fourteen electrodes from a 64-channel braincap (Easycap) (Fig. 1A: 8, 10, 12, 14, 16, 31, 33, 42, 49, 55, 56, 59, 62, and 63) were recorded for a parallel study. However, only four EEG recording electrodes (Fig. 1B: channel 56 and 62, right and left mastoids; channel 49, Inion; channel 59, ~3.5 cm below the Inion) were used for data analysis in this article, focusing on potential clinical applications (*Hu & Dietz, 2015*; *Hu, Kollmeier & Dietz, 2015*). The FPz served as the ground and Cz as the physical reference. Impedances were kept below 10 KΩ and checked after each run, adjusting if necessary. The scalp electrodes were connected to the 70-channel SynAmps RT amplifier system (Neuroscan) *via* monopolar input connectors. The voltage resolution was approximately 29.8 nV/LSB, and the recordings were filtered by an analog antialiasing lowpass filter with a corner frequency of 8 kHz, digitized with a 20 kHz *via* a 24-bit A/D converter.

## Test procedure

A flow chart of the performed experiments in this study was provided in Fig. 2. Before conducting the EEG experiments, each participant underwent approximately 30 min of

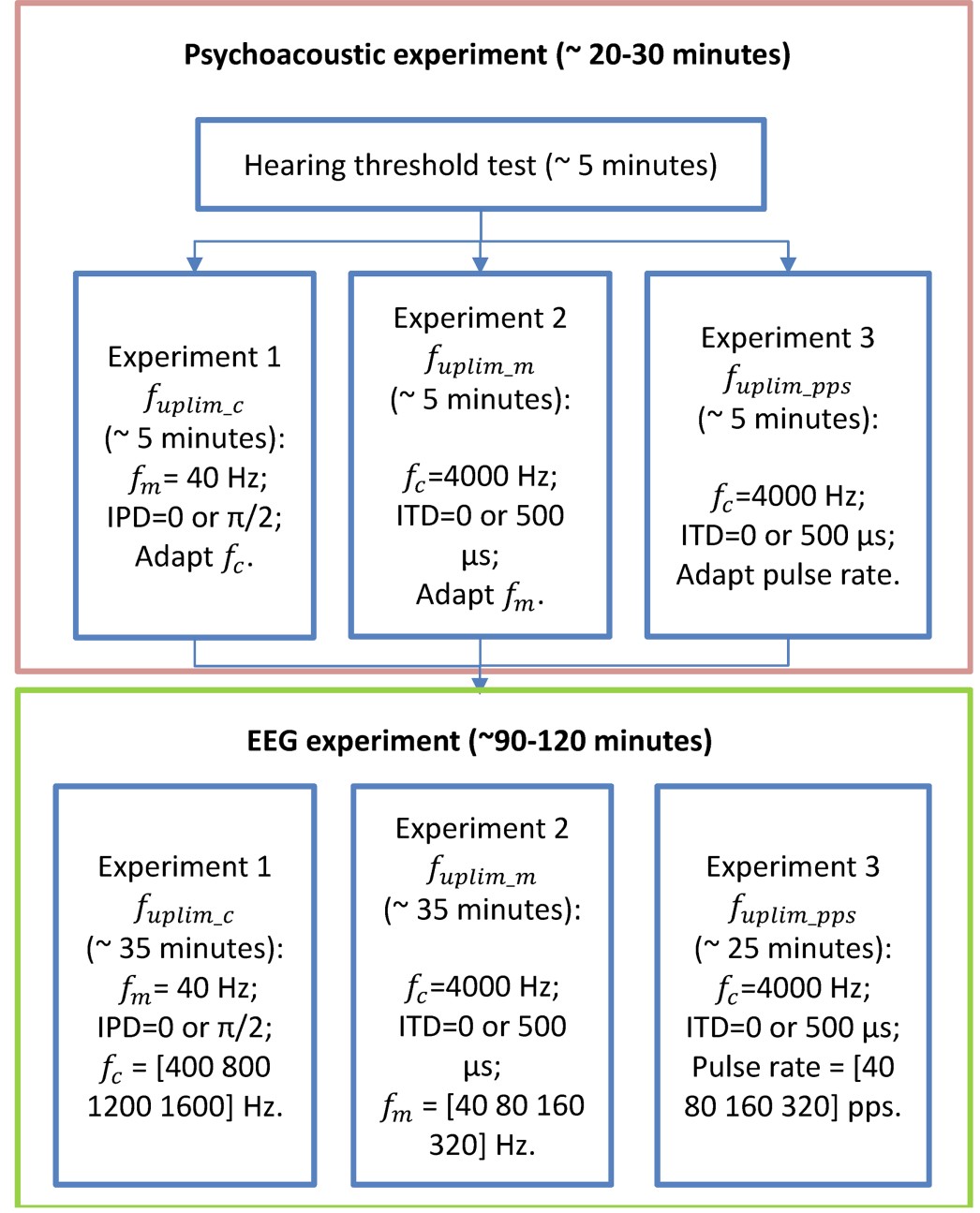

**Figure 2 Flow chart of the experimental stages.** The flow chart of the experimental stages. Participants first undergo listening experiments (enclosed in the orange box, including audiometry, and three lateralization experiments) and then three EEG experiments (enclosed in the green box). The order of the three experiments in both psychoacoustic and EEG experiments was randomized for each participant.

pretests on a separate day to assess their upper-frequency limits of $ITD_{FS}$ and $ITD_{ENV}$ sensitivities: 1) the upper limit of the carrier frequency ($f_{uplim\_c}$) for detecting changes of $ITD_{FS}$ in 40 Hz modulation frequency SAM tones with carrier frequency below 2,000 Hz; 2) the upper limit of the modulation rate ($f_{uplim\_m}$) for detecting changes of

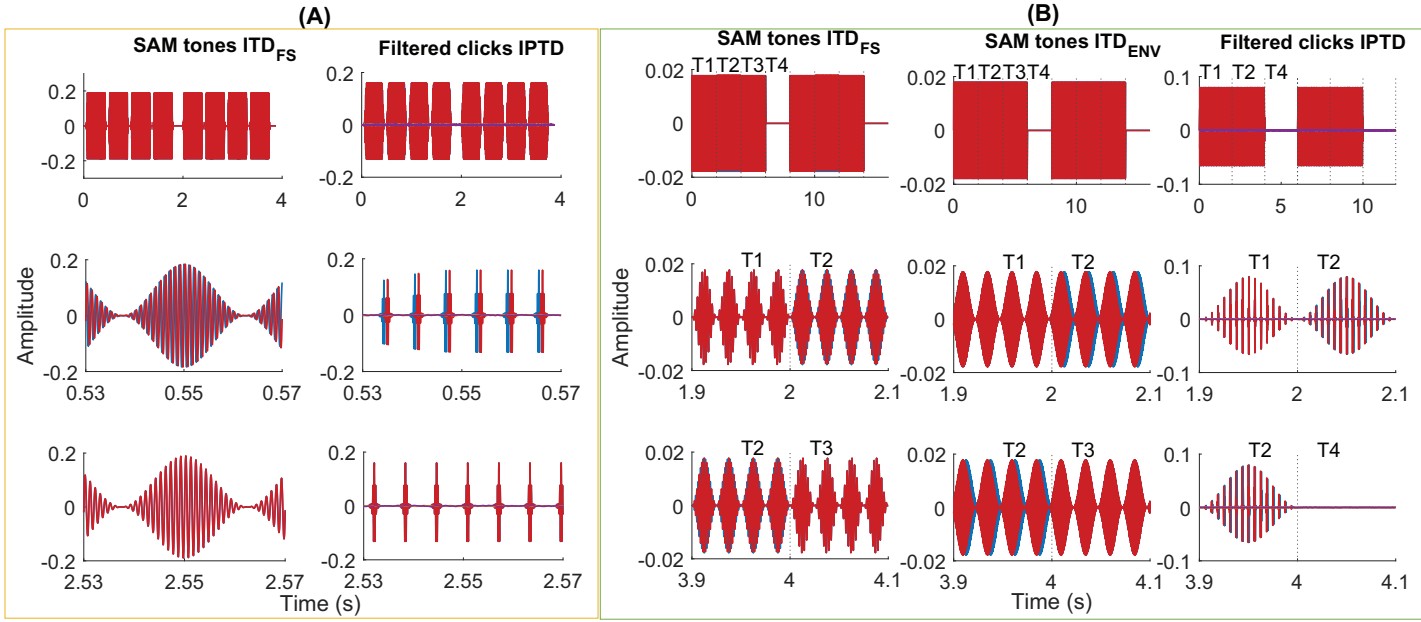

**Figure 3 Exemplary stimuli.** Exemplary stimuli used in the psychoacoustic (A) and EEG experiments (B). Two intervals of SAM stimuli (A, column 1, $f_m$ = 40 Hz, $f_c$ = 1,000 Hz, $ITD_{FS}$ = π/2) and filtered clicks (A, column 2, pulse rate = 160 pps, $f_m$ = 2.5 Hz, IPTD = 0 or IPTD = 500 μs) in the psychoacoustic experiments. Two consecutive intervals were separated by 500 ms of silence. Each interval contains four consecutive 400-ms tones (including 20-ms cosine rise/fall ramps), separated by 100 ms. In these examples, the first interval is the target, where the first and third tones were the same as in the standard interval while the second and fourth tones differed in their IPD by π/2 or by IPTD of 500 μs. Two repeats of the SAM stimuli with $ITD_{FS}$ (B, column 1, $f_m$ = 40 Hz, $f_c$ = 400 Hz, IPD = π/2), SAM stimuli with ITDenv (B, column 2, $f_m$ = 40 Hz, $f_c$ = 4,000 Hz, $ITD_{ENV}$ = 500 μs), and filtered clicks (B, column 3, pulse rate = 160 pps and $f_m$ = 10 Hz, IPTD = 0 or IPTD = 500 μs) were used in three EEG experiments. The duration of each presentation is 8 s (experiments 1 and 2) or 6 s (experiment 3). It includes 2 s of the diotic stimuli (time window T1), 2 s of the dichotic stimuli (time window T2; T→T2 named outward switching), 2 s of the standard stimuli (time window T3; T→T3 named inward switching; for experiment 1 and 2), and 2 s of silence (time window T4).

$ITD_{ENV}$ in SAM tones with carrier frequency of 4,000 Hz. 3) the upper limit of the pulse rate ($f_{uplim\_pps}$) for detecting interaural pulse time difference (IPTD) using band-pass filtered clicks. For more details regarding the "Psychoacoustic pretest-experiments" and their results, please refer to *Hu et al. (2023c)* and the Supplemental Materials.

The second appointment was dedicated to the three independent EEG experiments. Four of the 12 participants who attended the psychoacoustic pretests were unable to participate in the EEG appointment due to the outbreak of COVID-19. Each experiment consisted four runs of EEG recordings. Each run included three repetitions of four basic blocks. Each basic block comprised of a sequence of five repeated EEG stimuli of one test condition. The presentation order of these resulting 12 blocks was randomized in each run. In addition, the order of the 12 runs (3 experiments × 4 runs per experiment) was also randomized for each participant. In total, there were 60 repetitions of each test condition. It took approximately 35 min for experiment 1, 35 min for experiment 2, and 25 min for experiment 3. During the recording, any ongoing artifact rejection was turned off, and filtering, artifact analysis, and averaging were done offline.

## Stimuli

Figure 3 shows some exemplary stimuli used in the psychoacoustic pretest (A) and EEG experiments (B). Please refer to the Supplemental Materials for more details about the generation of test stimuli.

### Experiment 1

The EEG responses were measured as a function of carrier frequency, using SAM tones generated according to *Bernstein & Trahiotis (2012)*, *Hu, Klug & Dietz (2022)*. The modulation frequency was $f_m = 40$ Hz, and $\text{ITD}_{\text{FS}}$ was introduced by applying interaural phase differences (IPD) to the carrier of SAM tones. Four carrier frequencies, $f_c = [400, 800, 1,200, 1,600]$ Hz, were tested. The EEG stimuli lasted for 8 s (s) and followed a sequence: 2 s of the diotic stimulus (IPD = 0 in time window T1), 2 s of the dichotic stimulus (IPD = π/2 in time window T2; T1–>T2 referred to as outward switching), 2 s of the standard stimulus (IPD = 0 in time window T3; T2–>T3 referred to as inward switching), and 2 s of silence (in time window T4). An example of a stimulus used in EEG experiment 1 is illustrated in Fig. 3B, column 1 (40 Hz SAM tones with $\text{ITD}_{\text{FS}}$, $f_c = 400$ Hz and $f_m = 40$ Hz).

### Experiment 2

SAM tone at carrier frequency of $f_c = 4,000$ Hz and four modulation frequencies $f_m = [40, 80, 160, 320]$ Hz were tested, with the ITD applied to the envelope instead of the carrier. Note that, as in *Kohlrausch, Fassel & Dau (2000)*, no precautions were taken to mask possible distortion products in experiment 2. Since the modulation frequencies were below 320 Hz, we believe that the main results summarized below are not affected by detection cues based on distortion products.

Similar to the EEG paradigm in experiment 1, the EEG stimuli sequence consisted of 2 s of the diotic stimulus ($\text{ITD}_{\text{ENV}} = 0$ in the time window T1), 2 s of the dichotic stimulus ($\text{ITD}_{\text{ENV}} = 500$ μs in the time window T2; with an outward switching T1–>T2), 2 s of the standard stimulus ($\text{ITD}_{\text{ENV}} = 0$ in the time window T3; with an inward switching T2–>T3), and 2 s of silence (in the time window T4). An example of a stimulus used in EEG experiment 2 is presented in Fig. 3B, column 2 (SAM tones $\text{ITD}_{\text{ENV}}$, $f_c = 4,000$ Hz and $f_m = 40$ Hz).

### Experiment 3

To further simulate the CI performance in NH listeners, filtered click trains were generated as described in *Hu et al. (2017)*, *Hu, Klug & Dietz (2022)*, with the bandwidth limited to 3,000–5,000 Hz, and a center frequency of $f_c = 4,000$ Hz. A low-pass filtered noise, uncorrelated between the ears was added to the filtered click trains to mask potential distortion products. The low-pass noise was generated by creating broadband noise in the time domain, converting it to the frequency domain, and setting the power of all components above 1,000 Hz to zero. The noise was further manipulated to have a flat spectrum up to 200 Hz, with a decreasing spectral density of 3 dB/octave above 200 Hz. It was then filtered with a 5th-order, lowpass filter having a cut-off frequency of 1,000 Hz

(*Hu et al., 2017*), and gated with 50-ms raised cosine ramps. The test stimulus was centered within the noise presentation in time, which was presented at 40 dB SPL.

Four fixed pulse rates of [40, 80, 160, 320] pps were employed. Each EEG stimuli sequence had a duration of 6 s, consisting 2 s of the diotic stimulus, followed by 2 s of the dichotic stimulus (with a transition from T1–>T2, referred as outward switching), and 2 s of silence (in time window T4, with a T2–>T3 inward switching). Fig. 3B, column 3, provides an example of the stimuli used in the EEG experiment 3 (filtered clicks IPTD, pulse rate = 160 pps and $f_m$= 10 Hz, IPTD = 0 or IPTD = 500 μs).

### EEG data analysis

For the EEG results, continuous EEG data was collected from each participant and segmented into epochs over a window of 8.2 s (experiment 1 and 2) or 6.2 s (experiment 3), including a pre-stimulus duration of 200 ms. In total, there were 60 epochs for each test condition. The data were digitally filtered using a two-order Butterworth band-pass filter with a frequency range of 0.1–1,000 Hz following the segmentation. The baseline was corrected by the mean amplitude of the last 1 s after the stimulus offset time window, and epochs with voltages exceeded ± 200 μV were rejected from further analysis. Then, the EEG data was averaged separately for each condition in each participant. As Cz is a commonly used channel for cortical responses, the recordings in this study were re-referenced to the average of the four clinical recording channels (Fig. 1B, 56, 62, 49, and 59).

To obtain the transient response in the time domain, the responses were filtered further using a second-order Butterworth band-pass filter with a commonly used frequency range of 0.1–30 Hz (*e.g.*, *Papesh, Folmer & Gallun, 2017*; *Calcus, Undurraga & Vickers, 2022*). The peak amplitudes and latencies of the auditory-evoked P1, N1, and P2 for each participant, along with the average response across all participants, were automatically identified within fixed time windows (10–85 ms for P1, 85–160 ms for N1, and 160–300 ms for P2) after the onset, change, or offset of the stimuli, as described in *Calcus, Undurraga & Vickers (2022)*. To obtain the ASSRs in the frequency domain, the filtered data between 0.1–1,000 Hz was used. ASSRs within different time windows (T1, T2, T3, T4, and T1234 in EEG experiments 1 and 2; T1, T2, T4, and T124 in EEG experiment 3) were obtained, where T1234 and T124 refer to the whole duration of each stimulus (8 or 6 s).

To explore the time-frequency characteristics of the evoked responses, a complex Morlet wavelet $\omega$ defined as Eq. (1) (*Cohen, 2019*) was used for visualization, where $f$ is the frequency in Hz, $t = -1 : \dfrac{1}{f_s} : 1$ is the time in seconds, $f_s$ is the sampling rate, and $i = \sqrt{-1}$. $\sigma$ is the width of Gaussian, and $n$ is the number of cycles. Normally n is between 2 to 15 for neuro-electrical signals with frequencies between 2 and 80 Hz. In this study, $n = 6$ unless otherwise stated.

$$\omega = e^{2i\pi ft}e^{-\frac{t^2}{2\sigma^2}},$$

$$\sigma = n/(2\pi f) \tag{1}$$

## Statistical analysis

Statistical analyses were conducted using IBM SPSS (version 27, IBM Corp., Armonk, NY, USA). To assess the effect of stimulus type (*i.e.* carrier frequency in experiment 1, modulation rate in experiment 2, and pulse rates in experiment 3) and response type (onset CAEP; outward change response ACC1; inward change response ACC2; offset CAEP) on the amplitude (*e.g.*, N1P2, ASSR) and latency (*e.g.*, N1), a general linear model repeated-measures (GLMrm) analysis was performed. If the sphericity assumption was violated, a Greenhouse-Geisser correction was applied. If necessary, pairwise *post-hoc* comparison *t*-tests with Bonferroni correction were performed for further analysis. Bonferroni adjusted *p*-values are reported as provided by SPSS. Unless stated otherwise, 1) the significance threshold is $p < 0.05$; 2) the number of comparisons used for the adjustment by SPSS is the number of possible pair-wise comparisons for the assessed factor (*i.e.*, for a factor with two levels, no correction was applied; for the factors with three levels, the *p*-values were multiplied by three in the adjustment).

# RESULTS

## Psychoacoustic pretest results

The pretests confirmed that all eight participants in the EEG experiments did not show salient ITD sensitivity at the highest carrier frequency (1,600 Hz) used in experiment 1, the highest modulation frequency of 320 Hz used in experiment 2, and the highest pulse rate of 320 pps used in experiment 3. Therefore, we expected that there would not be any detectable ITD change responses at these upper limit conditions. Anecdotally, a non-experienced participant described the low-frequency SAM tones rhythmic, but the high-frequency stimuli as "annoying and like background noise".

## EEG results

### Time domain (CAEPs)

To validate our assumptions listed in the introduction (assumptions 1 to 4), we first examine the CAEPs in the time domain here. Figs. 4A–4C shows the individual (gray) and average EEG responses for experiments 1–3 in the time domain. The same processing procedure and automatic peak-detection method as described in "EEG data analysis" section were applied to all conditions. Each figure presents the four test conditions using different colors. The P1, N1, and P2 peaks of the average responses are marked with triangle symbols in each of the test conditions in panels 1–4. For easy comparison, the averaged data from panels 1–4 are overlaid in panel 5.

Figure 5 replots some results from Fig. 3. The upper panel shows the average responses evoked by 40 Hz SAM tones with $ITD_{FS}$ changes ($f_c = 400$ Hz, named, fc400ITDfs, red) and by 40, 80, 160, and 320 Hz SAM tones with $f_c = 4,000$ Hz and $ITD_{ENV}$ changes. The lower panel shows the responses evoked by 40, 80, 160, and 320 pps filtered click trains with interaural pulse time difference (IPTD) changes, in addition to the fc400ITDfs (red). By overlaying these curves, some ACC-like responses were able to be identified within the same ACC time duration as fc400ITDfs (after 2 and 4 s, red curve). However, the ACC responses evoked by envelope ITD changes were much smaller than those evoked by $ITD_{FS}$

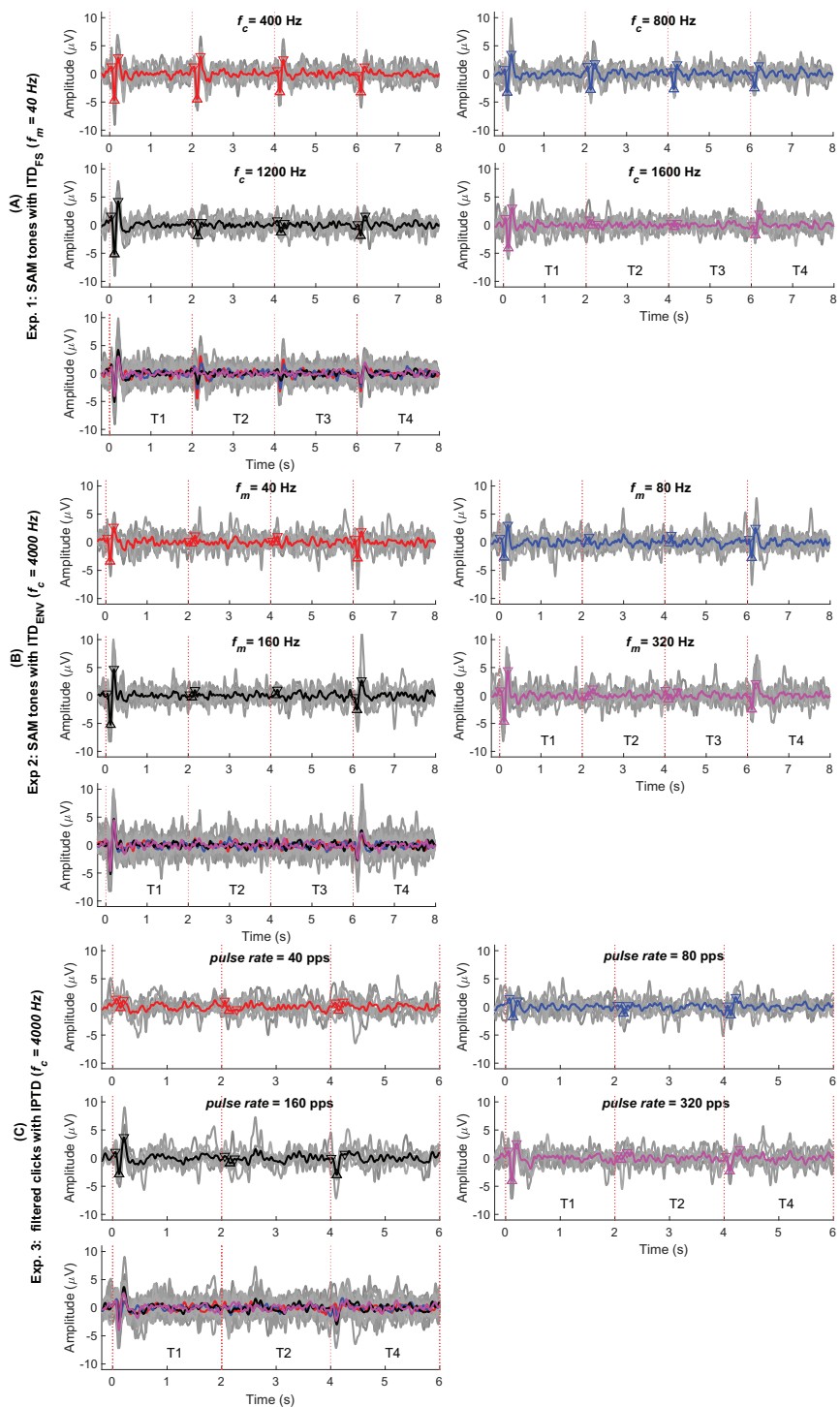

**Figure 4 Average response in the time domain.** The red, blue, black, and pink waveforms are the overall average CAEPs across participants for the four test conditions. For easy comparison, the data from panels 1–4 are overlaid in panel 5. The gray curves are the average CAEPs of each participant. The dotted red vertical lines are the start time of onset, ACC1, ACC2, and offset. T1, T2, T3, and T4 are the corresponding time duration. The four conditions are $f_c$ = (400, 800, 1,200, 1,600) Hz in (A), $f_m$ = (40, 80, 160, 320) Hz in (B) and pulse rate = (40, 80, 160, 320) pps in (C).

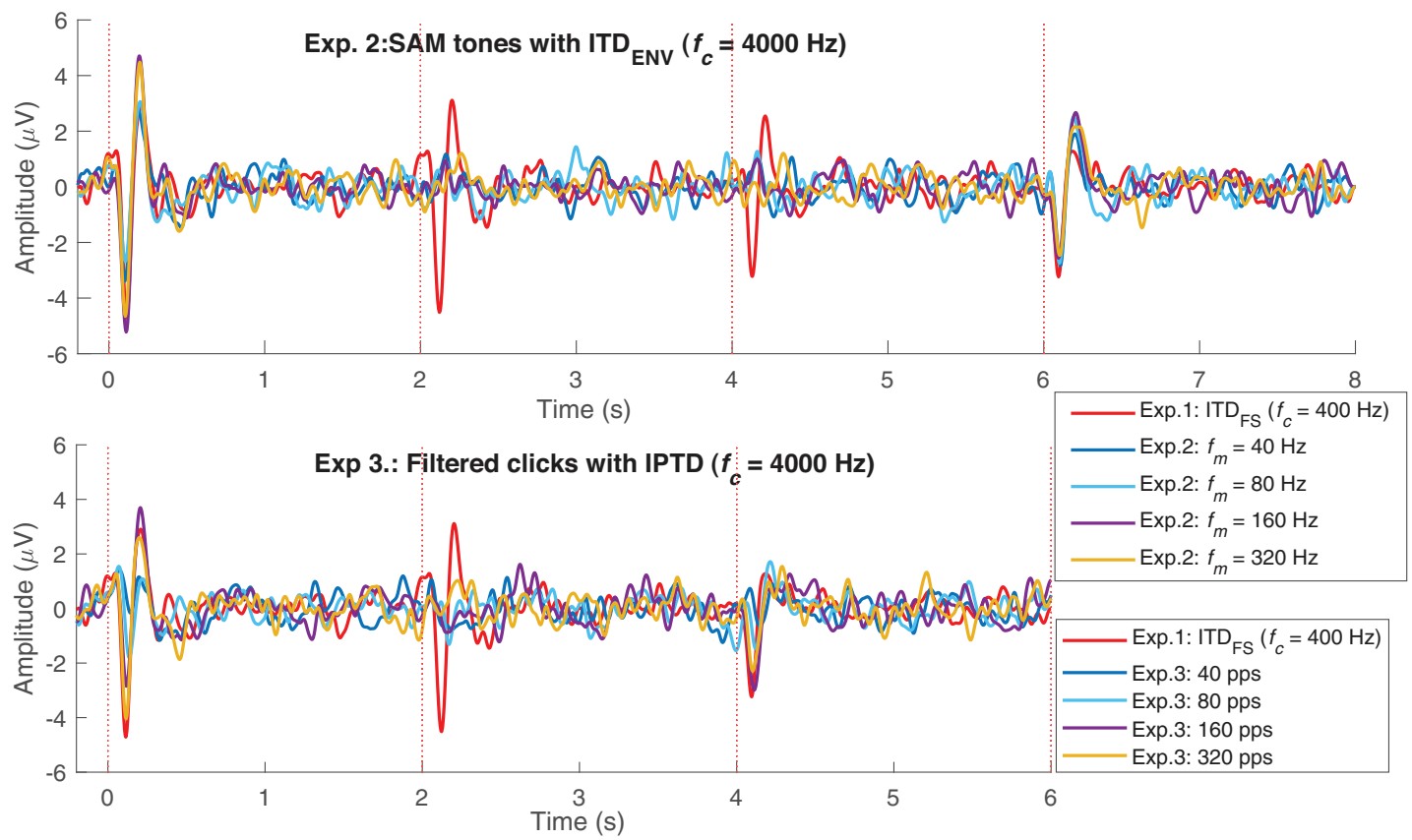

**Figure 5 EEG response overlay plot.** Overlay of average EEG responses for high-frequency SAM tones ($f_c$ = 4,000 Hz) in experiment 1 (Exp.1) and filtered clicks in experiment 2 (Exp.2) with modulation rate of 40, 80, 160, and 320 Hz or pps. The red curve represents the average response for 40 Hz SAM tones with $ITD_{FS}$ ($f_c$ = 400 Hz).

and close to the noise floor for both 4,000 Hz SAM tones (upper panel) and filtered clicks (lower panel). The automatically marked ACC peaks mostly fell within the same range as the noise within the detection windows in experiments 2 and 3, and the 1,600 Hz condition of experiment 1. Therefore, caution should be taken when interpreting the ACC peaks in these cases.

By observing the waveform morphology of the EEG responses shown in Figs. 4–5, and considering the information redundancy among P1, N1, and P2, in the statistical analysis, mainly the results of N1P2 amplitude (the amplitude difference between P2 and N1, *i.e.*, P2-N1, in μV) and the N1 latency were reported. Figs. 6A–6C shows the violin plots (*Hintze & Nelson, 1998*) of the automatically detected N1P2 amplitude and N1 latency of experiment 1–3. The pair-wise Bonferroni corrected *t*-tests with $p < 0.05$ were marked with '*' symbols.

In general, all test conditions in experiment 1 elicited both onset and offset CAEPs, confirming Assumption 1. Please refer to the following three subsections for more details.

● Assumptions 2: CAEPs in experiment 1

Experiment 1 was designed to validate the test setup and replicate results reported in the literature. Figs. 4 and 6A shows the amplitude and latency in experiment 1 ($ITD_{FS}$).

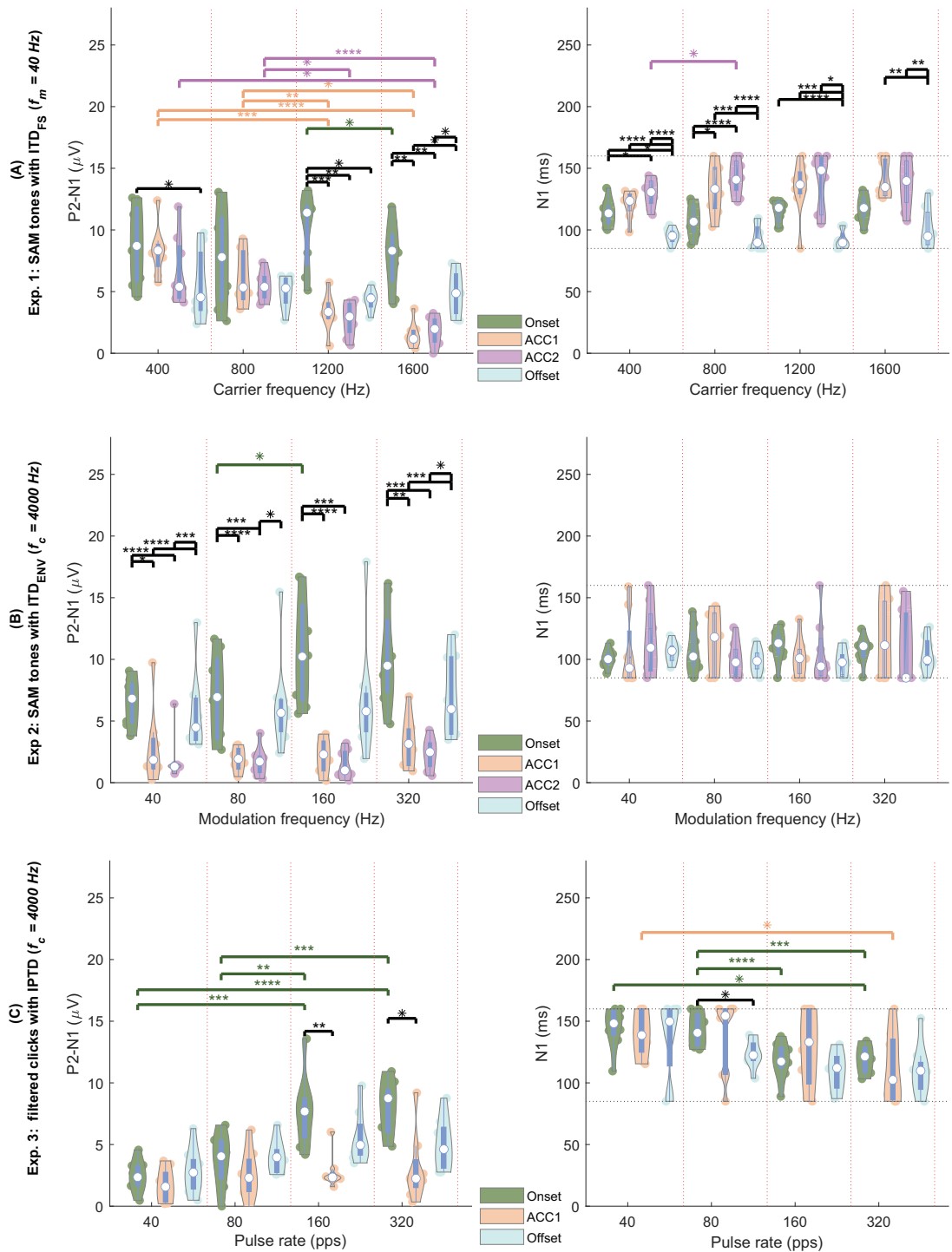

**Figure 6 EEG violin plot.** Violin plots of the N1P2 amplitude (left column, in μV) and the N1 latency (right column, in ms) of four response types (color coded) under four test conditions (separated by the vertical red dotted lines). (A) The four conditions are = (400, 800, 1,200, 1,600) Hz. (B) The four conditions are = (40, 80, 160, 320) Hz. (C) The four conditions are pulse rate = (40, 80, 160, 320) pps (A) carrier frequencies (400, 800, 1,200, 1,600 Hz, separated by the vertical red dotted lines). The solid colored dots within each violin plot are individual data from each participant. Each white dot represents the median. Conditions marked with asterisks (*; **; ***; ****) are significantly different ($p < 0.05$; $p < 0.01$; $p < 0.005$; $p < 0.001$, respectively). Black lines indicate comparisons between different response types within each frequency group, while different colors represent comparisons within a specific response type across frequencies.

The offset responses (aqua) were relatively consistent across different carrier frequencies and the N1 latency of the offset responses was generally shorter compared to the onset and ACC responses. In general, most results from experiment 1 were consistent with *Ross, Tremblay & Picton (2007)*, where CAEPS to IPD changes of SAM tones were recorded from 12 NH listeners using MEG. For example, the mean P1, N1, and P2 amplitudes of ACC were smaller than those of the onset response: P1, 1.684/1.405/1.005/ 0.248 μV; N1, 3.324/-1.299/-1.019/-2.146 μV; P2, 3.946/2.128/1.862/2.379 μV for onset/ ACC1/ACC2/offset. The ACC latencies were delayed compared with the corresponding onset and offset ones: P1, 42/46/57/27 ms; N1, 114/132/137/95 ms; P2, 211/227/240/213 ms for the onset/ACC1/ACC2/offset. The mean latencies of both P1 and N1 fell within a similar range, albeit slightly smaller than those reported by *Ross, Tremblay & Picton (2007)*. The latencies of $ITD_{FS}$ change evoked ACC1 and ACC2 were longer than the onset ones, however, the differences between ACC and onset responses were smaller than (*Ross, Tremblay & Picton, 2007*). In line with (*Ross, 2018*), a tendency for larger responses to outward IPD changes (ACC1) compared to inward changes (ACC2) was observed for the lower carrier frequencies. Nevertheless, this tendency did not reach statistical significance here ($p > 0.5$).

The N1P2 amplitude was significantly affected by carrier frequency ($f_c$), response type, and their interaction according to GLMrm ($p < 0.005$). There were no significant differences between 400, 800, and 1,200 Hz, but the N1P2 amplitude for the 1,600 Hz was significantly smaller than the other carrier frequencies. Pairwise comparisons showed no significant differences between carrier frequencies in most cases for both onset and offset CAEPs, except that the onset N1P2 amplitude of 1,200 Hz was slightly larger than that of 1,600 Hz ($p = 0.048$). For both ACC1 (outward) and ACC2 (inward) responses, there were no significant differences between 400 and 800 Hz, while the N1P2 amplitudes of 400 Hz and 800 Hz were significantly larger than those of 1,200 and 1,600 Hz. The onset CAEPs were significantly larger than the ACC1, ACC2, and offset CAEPs, but not significantly different amongst the other three types. Pairwise comparisons showed that the onset CAEPs were significantly larger than the offset CAEPs for 400 and 1,200 Hz, and no significant differences between ACC1 and ACC2 for all carrier frequencies. Significant correlations were observed between the onset N1P2 amplitudes of most carrier frequencies, except for 400 *vs* 1,200 Hz, and 400 *vs* 1,600 Hz.

The GLMrm analysis showed a significant effect of response type ($p < 0.001$) on the N1P2 latency, but no significant effect of $f_c$ and their interaction. The N1 latency of ACC responses was significantly larger than the onset response, while the offset response had the shortest latency and was significantly smaller than the other response types.

In summary: 1) Clear ACC responses were evident in experiment 1 for conditions with $ITD_{FS}$ changes in carrier frequencies up to 1,200 Hz, supporting Assumption 2. 2) The N1 latency of ACC responses was the longest, while offset responses had shorter N1 latency and smaller amplitude than both onset and ACC responses. 3) the N1P2 amplitudes of 400 Hz and 800 Hz were significantly larger than those of 1,200 and 1,600 Hz.

- Assumptions 3: CAEPs of experiment 2

In experiment 2 (ITD$_{ENV}$, Fig. 6B, also see Fig. 4), similar to experiment 1, there were clear onset and offset responses in all four test conditions. The onset N1P2 amplitude was comparatively larger than the offset responses, but the difference between the onset and offset CAEPs was smaller compared to those shown in Fig. 6A. Consistent with the findings of *Ross (2018)*, the N1P2 amplitudes of both onset and offset CAEPs were larger than the ACC responses, due to the tiny (close to the noise floor) or absence of ACC responses.

Regarding the N1P2 amplitude in experiment 2, a GLMrm analysis revealed significant effects of $f_m$, response type, and their interaction. The mean amplitude was 4.249/4.228/ 5.258/5.665 µV for $f_m$ of 40/80/160/320 Hz, respectively. A significant difference between $f_m$ was only observed for 80 Hz *vs* 160 Hz. Pairwise comparisons within each response type also only showed a just significant smaller onset N1P2 amplitude in 80 Hz condition compared to the 160 Hz condition ($p = 0.048$). The mean amplitude was 8.547/2.543/ 1.853/6.458 µV for onset/ACC1/ACC2/offset, respectively. Both onset and offset CAEPs were larger than the ACC responses. There were no significant differences between ACC1 and ACC2, and between onset and offset. Pairwise comparisons also showed no significant differences between them within each $f_m$. The onset and offset CAEPs were significantly larger than ACC responses, except for some cases with $f_m = 80$ Hz, and $f_m = 160$ Hz. Significant correlations were observed among modulation frequencies for all onset N1P2 amplitudes and for most offset N1P2 amplitudes, except for 320 *vs* 40, and 320 *vs* 160 Hz.

For N1 latency, the GLMrm analysis showed no significant effect of either $f_m$ or response type. The mean latency was 107/109/107/101 ms for onset/ACC1/ACC2/offset, and 107/105/104/109 ms for 40/80/160/320 Hz.

In summary, the ACC responses evoked by ITD$_{ENV}$ changes were much smaller than those evoked by ITD$_{FS}$ observed in experiment 1, supporting Assumption 3. However, categorizing ACC responses to ITD$_{ENV}$ changes as absent based solely on the signal-to-noise ratio might be misleading. For example, by overlaying multiple test conditions as shown in Fig. 5, some ACC-like responses became identifiable.

- Assumption 4: CAEPs of experiment 3

For the filtered click trains (Fig. 6C), no inward IPTD change (ACC2 responses) conditions were tested in experiment 3. Overall, the N1P2 amplitude of both onset and offset responses increased with increasing pulse rates. Similar to experiment 2, the ACC1 responses were either small (near the noise floor) or absent.

For N1P2 amplitude, GLMrm showed a significant effect of pulse rate, response type, and their interactions ($p < 0.01$). The mean amplitude was 2.31/3.43/5.36/5.35 µV for 40/ 80/160/320 pps, respectively. There were no significant differences between pulse rates of 40 and 80 pps, and between 160 and 320 pps. Within each response type, pairwise comparisons showed no significant differences between pulse rates for both ACC1 and offset responses. For the onset CAEPs, there were significant differences between most pulse rates ($p < 0.01$), except for conditions of 40 *vs* 80 pps, and 160 *vs* 320 pps. The mean amplitude was 5.48/2.52/4.34 µV for onset/ACC1/offset, and only the difference between

onset and ACC1 responses was significant. Within each pulse rate, pairwise comparisons showed no significant differences between response types for most pulse rates, except that for 160 pps and 320 pps, there was a significantly larger onset N1P2 amplitude than the offset one ($p = 0.009$, and $p = 0.014$). There were no correlations between N1P2 amplitudes of different pulse rates for both onset and offset responses.

For N1 latency, GLMrm revealed a significant effect of pulse rate, but not of response types or their interactions. The mean latency was 141/134/119/114 ms for 40/80/160/320 pps, respectively. The N1 latency was significantly shorter for 320 pps compared to 40 pps and 80 pps, and for 160 pps compared to 80 pps. The mean latency was 131/129/120 ms for onset/ACC1/offset responses with no significant differences between them.

In summary, we were unable to confirm Assumption 4, as ACC responses evoked by both ITD$_{ENV}$ changes in experiment 2 and IPTD changes in experiment 3 were very small and closely approached to the noise floor, suggesting their potential absent in the classical EEG response interpretation.

### Assumption 5: Frequency domain (ASSRs)

Figure 7 shows the average ASSRs across participants within the analysis window of T1234 or T124. Within each panel, the colored curves in the shaded area are the average ASSRs for each individual. The red, blue, black, and pink spectra are the overall average ASSRs across participants for different test conditions: (A) $f_c$ = [400, 800, 1,200, 1,600] Hz; (B) $f_m$ = [40, 80, 160, 320] Hz; (C) pulse rate = [40, 80, 160, 320] pps. The numbers with the same colors in the figure represent the ASSR amplitudes at the modulation frequency. The right panel displays the violin plots of the ASSR amplitude at the modulation frequency, within a duration of 8s (T1234) for the SAM tones or 6 s (T124) for the filtered clicks.

For experiment 1 (Fig. 7A), the group mean 40-Hz ASSR amplitudes were 0.187/0.173/0.152/0.142 µV for 400/800/1,200/1,600 Hz, respectively. The amplitude of the 40-Hz ASSR decrease gradually with increasing carrier frequency for the SAM-type stimuli, as previously reported by *Ross (2018)*. However, these differences are not statistically significant. The 40-Hz ASSR of SAM tones with different carrier frequencies were correlated, except for the comparison between 800 and 1,600 Hz.

Figures 7B and 7C show the ASSRs at different modulation rates (40, 80, 160, and 320 Hz). The numbers shown in different colors are the average ASSR values for each modulation rate. In general, the ASSRs for high carrier frequency stimuli (Fig. 7B, with a smaller y-axis scale) were smaller than those for low-frequency (<= 1,600 Hz) SAM tones (Fig. 7A), and the ASSRs for filtered clicks (Fig. 7C) were larger than those for high-frequency SAM tones. In both types of high-frequency stimuli, the order was 40 Hz ASSR > 160 Hz ASSR > 80 Hz ASSR > 320 Hz ASSR. The filtered clicks elicited larger ASSRs, as suggested by *Hu, Klug & Dietz (2022)*.

The mean overall amplitudes of the ASSRs in experiment 2 (Fig. 7B) were 0.093, 0.058, 0.071, and 0.022 µV for 40, 80, 160, and 320 Hz, respectively. The 320 Hz ASSR was significantly smaller than the others. There were no significant correlations between the ASSRs of different modulation frequencies, except for 160 Hz and 320 Hz.
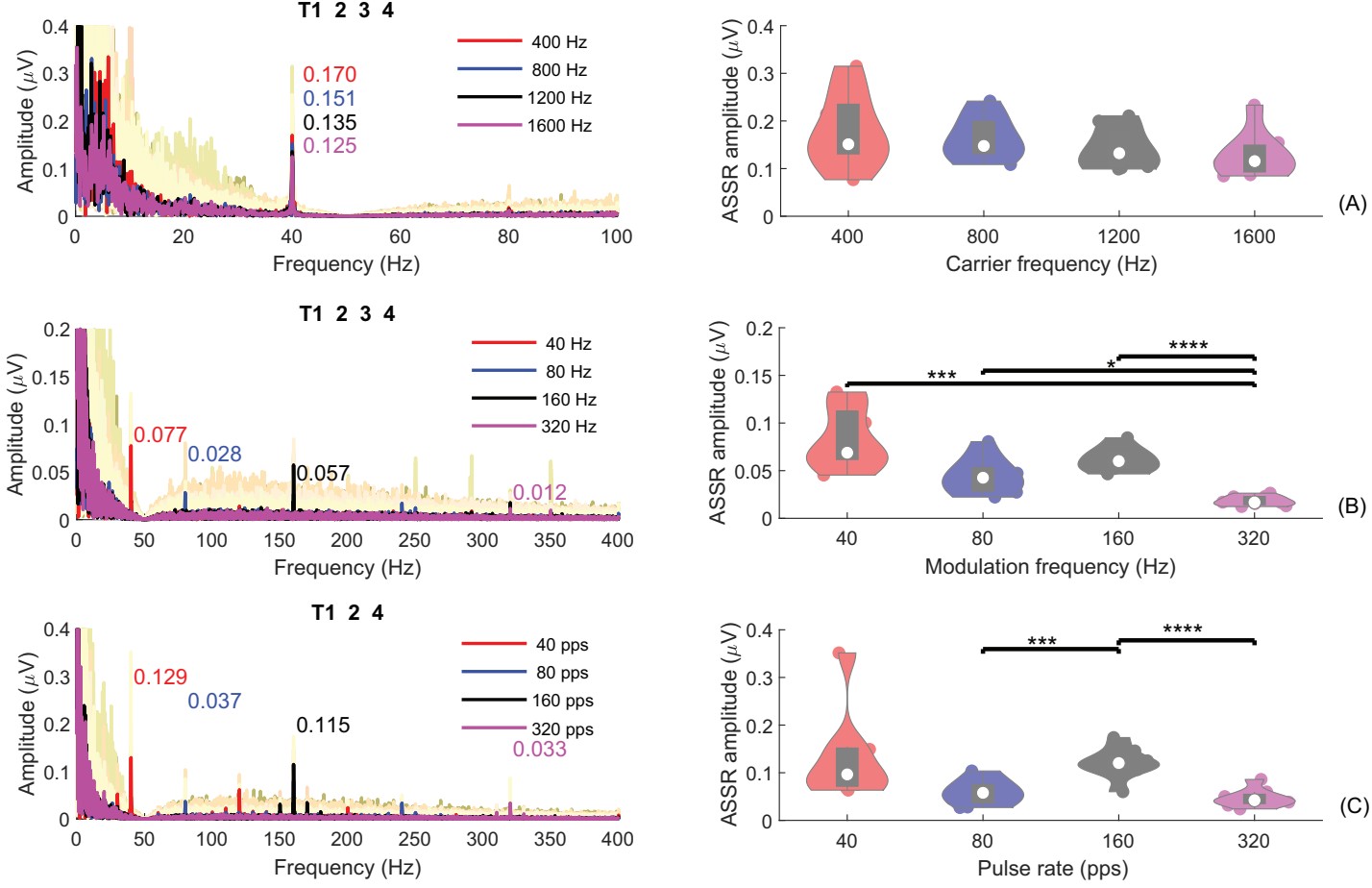

**Figure 7 ASSR plot.** The individual and group average ASSRs in the frequency domain, with analysis time window T1234, or T124. The group average ASSRs across participants are depicted in red, blue, black, and pink for different test conditions: (A) $f_c$ = (400, 800, 1,200, 1,600) Hz; (B) $f_m$ = (40, 80, 160, 320) Hz; (C) pulse rate = (40, 80, 160, 320) pps. The cream colored background curves are the average ASSRs for each individual. The numbers in different colors indicate the ASSR values at the modulation rate for different test conditions. The violin plots on the right display the corresponding ASSR amplitudes within an analysis window of 8 s (T1234) for SAM tones or 6 s (T124) for filtered clicks. The solid dots in each violin plot represent individual ASSRs at the corresponding or pulse rate for each participant. Conditions marked with asterisks (*; ***; ****) are significantly different ($p < 0.05$; $p < 0.005$; $p < 0.001$, respectively).

The overall mean amplitudes of the ASSRs in experiment 3 (Fig. 7C) were 0.148/0.075/0.123/0.048 μV for 40/80/160/320 pps, respectively. The 160 pps showed significantly larger ASSRs compared to 80 and 320 pps. Significant correlations were also observed between the ASSR amplitudes of 160 pps and 80 pps, 160 pps and 320 pps, and 320 pps and 80 pps.

Some studies suggested that the ASSRs for modulation frequencies up to 50 Hz are most likely generated from the auditory cortex (*Mäkelä & Hari, 1987*; *Pantev et al., 1994*; *Herdman et al., 2002*). To compare the 40-Hz ASSRs evoked by the 40-Hz modulated SAM tones and 40-pps filtered clicks within different analysis windows (see "Fig. S3"), GLMrm (with factors: stimuli type [400/800/1200/1600/4000SAM/40-pps-clicks], and analysis window [T1, T2, and T4]) showed a significant effect of stimuli type, analysis window ($p < 0.001$), and their interaction. The mean amplitudes for 400, 800, 1,200, 1,600, 4,000 Hz

SAM tones, and 40-pps filtered clicks were 0.168, 0.461, 0.142, 0.136, 0.091, and 0.154 µV, respectively. Pairwise comparison revealed significantly smaller amplitude of the 40-Hz ASSR evoked by the 4,000 Hz SAM tones compared to the four low-frequency SAM tones, but no significant differences between the other stimulus types, including 4,000 Hz SAM tones *vs* 40-pps filtered clicks. No significant differences were found between analysis windows T1 and T2, but as expected, T4 (silence) was significantly smaller than T1 and T2.

In summary, ASSRs were detectable in all three experiments, particularly for conditions with modulation rates below 320 Hz. ASSRs amplitudes are influenced by the modulation frequency, carrier frequency, and the type of stimuli used. Under the same modulation frequency, ASSRs were larger for conditions with carrier frequencies up to 1,600 Hz in experiment 1 compared to those with a carrier frequency of 4,000 Hz in experiments 2 and 3. This confirms the assumption 5. In addition, the ASSRs for filtered clicks were larger than those for high-frequency SAM tones. In both types of high-frequency stimuli, the order was 40 Hz ASSR > 160 Hz ASSR > 80 Hz ASSR > 320 Hz ASSR.

### Assumption 6: time-frequency domain

As shown in Fig. 5, it is more challenging to determine the presence of detectable ACC responses evoked by $ITD_{ENV}$ changes in high-frequency SAM tones and filtered clicks compared to those elicited by $ITD_{FS}$ changes in low-carrier frequencies. This requires more experience and possibly additional references, such as the overlying method demonstrated in Fig. 5. To gain a better understanding of the smaller ACC responses evoked by $ITD_{ENV}$ changes and to enhance data visualization, Fig. 8 displays the responses (with a higher cutoff frequency of 1,000 Hz instead of 30 Hz) in the time-frequency domain: SAM $ITD_{FS}$ (A), SAM $ITD_{ENV}$ (B), filtered clicks IPTD (C). The average response plotted in each subplot as the black solid line around ~10 Hz. In general, the time-frequency amplitudes are related to stimulus parameters, such as $f_m$, and $f_c$.

The onset and offset responses in the time-frequency domain (Fig. 8) displayed clear clusters of higher energy in the lower frequency range (<30 Hz) whenever there were detectable responses. In general, all three stimuli types evoked noticeable onset and offset CAEPs, except for the 40 pps filtered clicks.

The ACC responses evoked by $ITD_{FS}$ changes were easily recognizable in both response waveforms (Fig. 4A) and the time-frequency domain (Fig. 8A) for the 400, 800, and 1,200 Hz SAM. However, it was smaller in 1,200 Hz SAM and not distinguishable in the 1,600 Hz in both the time and time-frequency domains.

In general, the ACC responses elicited by $ITD_{ENV}$ changes were smaller compared to those evoked by $ITD_{FS}$ changes. It appears there is greater presence of low-frequency brain activity (<10 Hz) for both types of high-carrier frequency stimuli in some cases. The ACC responses are roughly similar in scale to neighboring brain activities, making it challenging to distinguish an ACC response even in the time-frequency domain (Figs. 8B and 8C).

As expected, Fig. 8 reveals 40-Hz ASSRs (between the two parallel red dashed lines around the 40 Hz) except during the 2s-silence period. Time-frequency visualization shows possible interactions between the transient CAEPs and the ASSRs. Whenever the P1-N1-P2 complex was detectable and prominent, there was a noticeable reset

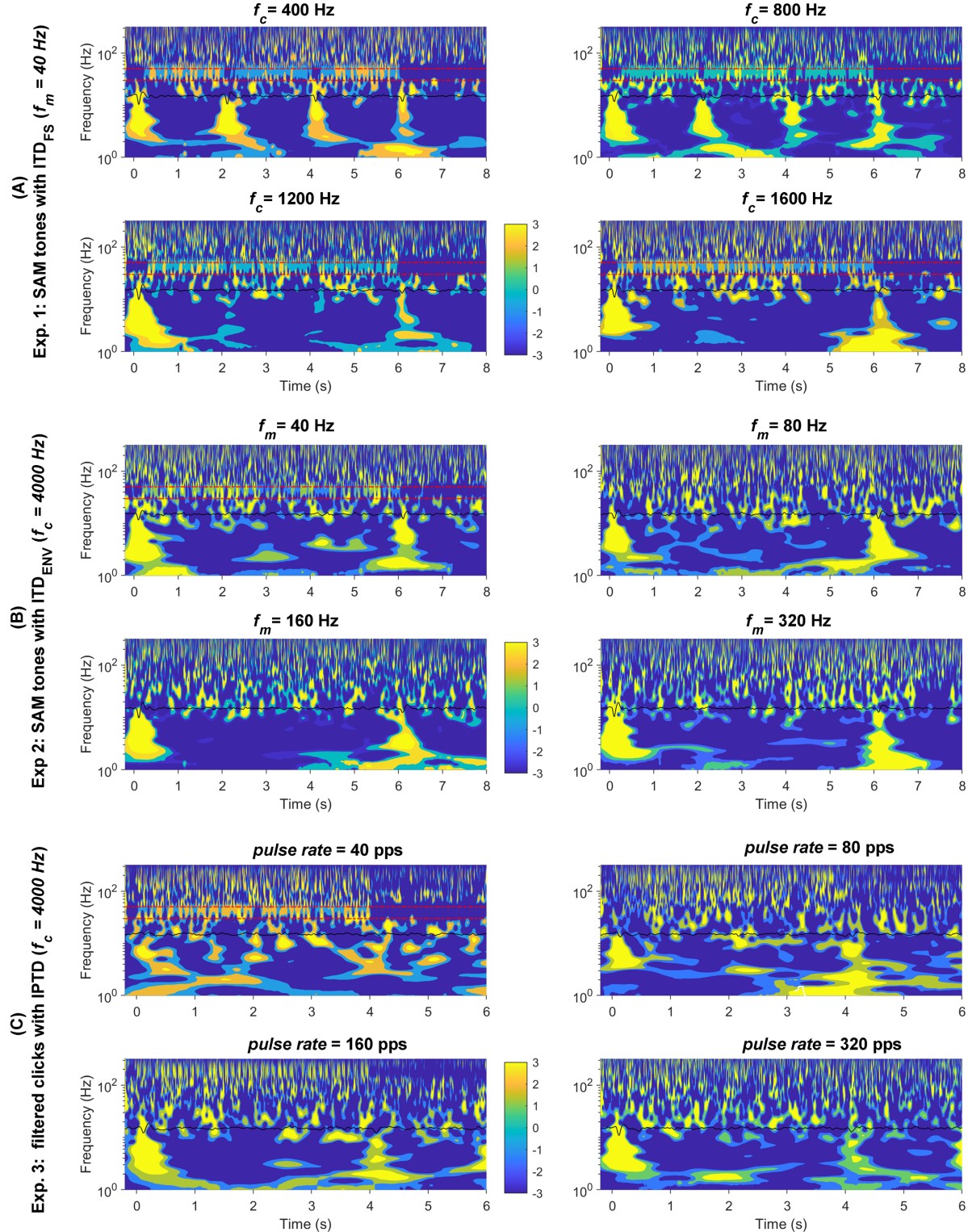

**Figure 8 Wavelet plot.** The average response in the time-frequency domain. The time-frequency spectrum was obtained through wavelet analysis of the average response, the black solid line shown in the center of each panel. The four conditions are shown in the upper left, upper right; bottom left, and bottom right panels: (A) $f_c$ = (400, 800, 1,200, 1,600) Hz;. (B) $f_m$ = (40, 80, 160, 320) Hz. (C) pulse rate = (40, 80, 160, 320) pps. The two parallel red dashed lines correspond to 30 and 50 Hz.

(represented by the blue gaps in the 40 Hz regions) in the ASSRs. This suggests the steady-state activity was desynchronized by the prominent transient CAEPs, which might similar to the findings reported as 'interrupt' responses by some other groups (*e.g.*, *Bianco et al., 2020*). For example, for $f_c$ of 400, 800, and 1,200 Hz, the ASSRs were suppressed or reset around 0, 2, and 4 s, respectively. There were notable energy differences in the ASSRs in these time windows (T1, T2, T3), as shown in Fig. S2 within the frequency range of 2–50 Hz.

In summary, the wavelet time-frequency analysis indeed provided us with additional insights into the data. For example, it suggests potential interactions between the transient CAEPs and the ASSRs, confirming our assumption 6.

## DISCUSSION

It has been reported in the literature that fine structure IPD- or ITD-based discrimination could be an important indicator for the performance in sound localization and the perception of speech related to phase locking in the auditory pathway (*Ross et al., 2007*; *Ross, Tremblay & Picton, 2007*; *Füllgrabe, Moore & Stone, 2015*; *Papesh, Folmer & Gallun, 2017*; *Eddins & Eddins, 2018*; *Eddins, Ozmeral & Eddins, 2018*; *Koerner et al., 2020*; *Moore, 2021*). In this study, CAEPs to ITD changes using an ACC paradigm was selected as a diagnostic assessment of binaural temporal processing. This paradigm has been demonstrated as robust and easily detectable in populations with various hearing profiles, both monaurally (*Mathew et al., 2017*, *2018*; *Han & Dimitrijevic, 2020*; *McGuire et al., 2021*; *Undurraga et al., 2021*; *Calcus, Undurraga & Vickers, 2022*) and binaurally (*Ross et al., 2007*; *Papesh, Folmer & Gallun, 2017*). Here, the ACC paradigm was simplified and recorded using a clinic-friendly EEG setup that can be conducted in less than 1 h. Multiple responses were recorded simultaneously within the same EEG paradigm and analyzed. The same paradigm was used to compare the ITD sensitivity between low-frequency stimuli with $ITD_{FS}$ and high-frequency stimuli with $ITD_{ENV}$. The EEG responses at different carrier frequencies and modulation rates were collected from the same participants within the same session. The response morphology of NH young participants was characterized as benchmark data for future studies involving the aging population and bilateral CI users.

In experiment 1 ($ITD_{FS}$), 40 Hz modulated SAM tones with carrier frequencies of 1,600 Hz or less were employed. The ACC evoked by IPD changes and the CAEPs evoked by the onset and offset of the SAM tones were recorded during the same session. Overall, the onset CAEPs and the ACC displayed similar morphologies. However, the mean peak latencies of ACC were generally longer than those of the onset and offset CAEPs (P1, 42/46/57/27 ms; N1, 114/132/137/95 ms; P2, 211/227/240/213 ms for onset/ACC1/ACC2/offset). Since the IPD change was introduced at the trough of the stimuli, the effect of a new stimuli onset on the ACC response were minimized. Consequently, the present data suggested that the ACC responses were evoked by acoustic changes in the ongoing stimuli (*e.g.*, IPD changes in experiment 1) rather than the onset of new stimuli. This supports the notion that the ACC is more than a simple onset response (*Ostroff, Martin & Boothroyd, 1998*). However, the ACC exhibits differences and similarities with the onset and offset

CAEP, indicating that these three responses may involve different but overlapping neural mechanisms.

In addition to the aforementioned three types of CAEPs, the ASSRs in the frequency domain were derived from the same data. The amplitude of the 40-Hz ASSRs gradually decreased with increasing carrier frequency, consistent with *Ross, Tremblay & Picton (2007)*, *Ross (2018)*. The mean ASSR amplitude was 0.187/0.173/0.152/0.142 for 400/800/ 1,200/1,600 Hz, respectively. The wavelet-based time-frequency visualization showed the interaction between detectable transient CAEPs and the steady state responses, particularly for the lower carrier frequency SAM tones (*e.g.*, 400, 800, and 1,200 Hz).

In summary, the method described in experiment 1 has potential as a tool for objectively evaluating the processing of changes in binaural information, especially for binaural temporal processing abilities. However, future research is needed to assess its variability (test-retest reliability) and relationship to behavioral tasks. To determine the test conditions for the EEG experiments in this study, we measured the ITD sensitivity of each participant around their upper limit frequency in the pretests, using an adaptive threshold measurement procedure (see Supplemental Materials). To investigate the possible correlation between the EEG findings reported here and the corresponding behavioral performance, a more appropriate measure would have been the ITD sensitivity in percentage correct rate. Despite this, the results are encouraging and represent a step towards an objective measure that could be used to study binaural cue processing in a large population within a half-hour session. In clinical applications, the test conditions could be further reduced to shorten the required measurement time. The 2 s stimuli was recommended for optimal clinical application of the ACC paradigm to provide a more robust response, which is consistent with the 1–2 s longer stimuli recommended by *Mathew (2018)* and *Calcus, Undurraga & Vickers (2022)*. There is a time trade-off because fewer epochs would be possible in the same recording time.

A sample rate of 20 kHz was used in the EEG recording, which is much higher than the typical 1–2 kHz used in classic cortical EEG paradigms. In the post-processing, the EEG data was downsampled to various rates between 1–20 kHz to assess the minimum required sample rate for detecting transient CAEPs and ASSR. Although the average data was not significantly impacted by downsampling, the individual data showed differences between 1 kHz and higher rates (*e.g.* 8 and 20 kHz). Based on the results, a 4-channel EEG setup with a sampling rate above 1 kHz was recommended for clinical use in acoustic hearing assessments. If time and storage limitations are not an issue, higher channel numbers (*e.g.*, 32 or 64 channels) and higher sampling rates (>2 kHz) could offer the opportunity to employ more advanced post-processing techniques. However, further optimization may be required for electrical hearing assessments in CI users.

In experiments 2 and 3, high carrier frequency SAM tones and bandpass filtered clicks were used, respectively, to study the potential differences in ACC morphologies evoked by $ITD_{ENV}$ changes compared to those evoked by $ITD_{FS}$ changes in NH participants. The use of bandpass filtered click trains was intended to mimic CI pulsatile stimulation and establish a foundation for future studies on CI users. The center frequency of 4,000 Hz was

chosen to eliminate or reduce phase locking in the fine structure, as $ITD_{ENV}$ changes with low carrier frequencies (<2,000 Hz) can primarily be attributed to fine structure changes.

Most participants in *Ross (2018)* showed no ACC responses to $ITD_{ENV}$ changes for the 4,000 Hz $f_c$ SAM tones. Our findings confirmed Ross' results that the majority of ACC responses were at the same level as the noise floor for $f_c$ of 4,000 Hz, making it challenging to detect ACC responses. Upon careful examination of overlaid plots (*e.g.*, Fig. 5) depicting different test conditions, we cautiously suggest the potential presence of ACC responses for modulation frequencies of 40, 80, and 160 Hz. Similarly, modest ACC responses were observed for the bandpass-filtered clicks. This suggests that the application of this paradigm to CI users is feasible, though it may pose difficulties, particularly due to the presence of CI stimulation artifacts (*Hu & Dietz, 2015*; *Hu & Dietz, 2015*; *Hu & Ewert, 2021*; *Hu, Williges & Vickers, 2023d*). However, CI users may exhibit larger ACC responses than NH participants due to direct electrical stimulation of the auditory nerve. For example, clear electrically evoked brainstem binaural interaction component (ABR-BIC) has been recorded in bilaterally implanted animals (*e.g.*, in cat, *Smith & Delgutte, 2007*; *Hancock et al., 2010*) and human CI users (*Pelizzone, Kasper & Montandon, 1990*; *He, Brown & Abbas, 2010*; *Gordon et al., 2012*; *Hu & Dietz, 2015*; *Hu, Kollmeier & Dietz, 2016*), whereas acoustic ABR-BIC results are mixed. Some authors have reported an inability to measure acoustic BIC (*Haywood et al., 2015*), while others suggested it as a potential tool for binaural hearing tests (*e.g.*, *Ungan, Yağcıoğlu & Özmen, 1997*; *Riedel & Kollmeier, 2002*; *Benichoux et al., 2018*; *Brown et al., 2019*).

The ASSRs in experiments 2 and 3 were generally smaller than in experiment 1. The filtered clicks evoked larger ASSRs than high-frequency SAM tones, with the order of 40 Hz>160 Hz >80 Hz >320 Hz for both stimuli. The mean ASSR amplitudes for the 4,000-Hz SAM tones modulated at 40/80/160/320 Hz were 0.093/0.058/0.071/0.022 μV, and for filtered clicks at 40/80/160/320 pps, they were 0.148/0.075/0.123/0.048 μV. The results suggest that the ability exists to follow the high-frequency stimulus envelope at cortical or subcortical level, but the encoding of envelope information may differ between high and low frequency stimuli. The low-frequency $ITD_{FS}$ evoked higher ACC responses than the high-frequency $ITD_{ENV}$ (*e.g.*, Fig. 8), which also suggests potential differences in the central part. In a recent article, Oxenham speculated that the human auditory cortex may process lower frequencies more effectively (*Verschooten et al., 2019*).

## Conclusion and implications of possible applications

The findings of this study have several important implications. Firstly, ACC responses can be evoked by $ITD_{FS}$ changes in the ongoing stimuli. They can be recorded without participant involvement and are easily detectable with the 4-channel EEG setup. This has the potential as an objective tool for evaluating binaural sensitivity or for documenting binaural training effects. Secondly, the 4-channel clinical EEG setup that we used included an automatic P1, N1, and P2 peak-picking classifier, that worked well within the defined time window. This is promising and only requires a sampling rate of greater than 1 kHz, making it clinically viable. Thirdly, the small ACC responses evoked by the envelope ITD may pose a challenge when applying this paradigm to bilateral CI users, particularly in the

presence of CI stimulation artifacts. However, these responses may be clearer in CI than in NH participants (*e.g.*, *Hu & Dietz, 2015*), thus may actually be much easier to detect. Lastly, our research team plan to test this paradigm in various populations with different hearing profiles to investigate differences in the neural encoding and processing of binaural and spatial cues. Further research is needed to characterize different types of CAEPs and ASSRs in CI users or participants with different degrees of neural deficits and to further develop the methods as a tool for remediating spatial processing deficits in these groups.

## ACKNOWLEDGEMENTS

The views expressed are those of the author(s) and not necessarily those of the MRC, NIHR or the Department of Health and Social Care. The authors are grateful to Brian Moore for sharing their TFS-AF test and to all the participants.

### Funding

This work was funded by a Medical Research Council Senior Fellowship grant (MR/S002537/1), EMSATON (Projektnummer 415895050), and the National Institute of Health Research (NIHR) Programme Grant for Applied Research (201608) and Biomedical Research Centre (203312). The funders had no role in study design, data collection and analysis, decision to publish, or preparation of the manuscript.

### Grant Disclosures

The following grant information was disclosed by the authors:
Medical Research Council Senior Fellowship grant: MR/S002537/1, EMSATON (Projektnummer 415895050).
National Institute of Health Research (NIHR) Programme Grant for Applied Research: 201608.
Biomedical Research Centre: 203312.

### Competing Interests

The authors declare that they have no competing interests.

### Author Contributions

- Hongmei Hu conceived and designed the experiments, performed the experiments, analyzed the data, prepared figures and/or tables, authored or reviewed drafts of the article, and approved the final draft.
- Stephan D. Ewert conceived and designed the experiments, analyzed the data, authored or reviewed drafts of the article, and approved the final draft.
- Birger Kollmeier analyzed the data, authored or reviewed drafts of the article, and approved the final draft.
- Deborah Vickers conceived and designed the experiments, analyzed the data, authored or reviewed drafts of the article, and approved the final draft.

## Human Ethics

The following information was supplied relating to ethical approvals (*i.e.*, approving body and any reference numbers):

The study has the approval of the Ethics Committee of the University of Oldenburg (Drs.EK/2019/075).

## Data Availability

The behavior measurements and more detailed analysis are available in the Supplemental Files.

## Supplemental Information

Supplemental information for this article can be found online at http://dx.doi.org/10.7717/peerj.17104#supplemental-information.

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
