# Peer review of "Rate dependent neural responses of interaural-time-difference cues in fine-structure and envelope"

_PeerJ, doi:10.7717/peerj.17104_

## Round 0.1 · original submission · Major Revisions

Dear Authors, Major revisions are needed for this manuscript. Please refer to the comments from the two peer reviewers.

Reviewer 1 ·

Basic reporting

The introduction did not provide adequate rationales for several aspects of the experimental design:
1) Why use normal-hearing subjects instead of actual cochlear-implant subjects? The authors mentioned the issue of CI artifacts. However, that was not an adequate rationale, considering researchers have developed ways to minimize CI artifacts.
2) Why focus on fine structure and envelope ITD? There is relevant information in the first paragraph of the introduction. However, it should be stated more clearly. A definition of the terms would also benefit readers.
3) Why choose the three EEG measures? Why simultaneously record them? The authors should expand the literature review on binaural EEG and MEG studies. Otherwise, it is not easy to appreciate the necessity, significance, and novelty of using these EEG measures.
4) The authors should provide hypotheses or predictions regarding the study findings re: the experimental design.

Experimental design

A major issue is the power of the study. The authors should state the rationale for choosing a small sample size of 8.

Minor issue
The choice of EEG parameters. It is unclear why the 14 electrodes were chosen for the study. It is unclear whether only Cz or more electrodes were used for analysis. It is unclear why to use 10-85, 85-160, and 160-300ms as time windows to identify peaks.

Validity of the findings

Major issue:
1) As pointed out by the authors, ACC peaks could not be reliably identified from all subjects and conditions. Thus, the current analysis methods on the ACC responses may not be valid.
2) The study generated rich data; however, the current result section was difficult to read. It was challenging to grasp the main results in relation to the study questions. I strongly recommend that the authors simplify the results to highlight the main findings (e.g., with subheadings or a summary paragraph), guided by their study questions.

Minor issues:
1) I think it would help the readers if supplementary Figures 1 and 2 were moved to the main text.
2) Figures 3 and 5 are overwhelming. Please simplify or rearrange. Adding information about the conditions directly to the figure may help.

Additional comments

Jargons or terms are throughout the manuscript without a clear explanation/definition, e.g., line 110, ‘multi-information analyzing techniques’; line 128, ‘alternative forced choice framework’; line 186, “the same paradigm”; line 214, ‘a clinical paradigm’. Please check the whole manuscript.

Reviewer 2 ·

Basic reporting

Here the authors investigated EEG paradigms to record neural responses of ITD cues in human subjects with the ultimate goal of utilizing EEG to improve spatial hearing abilities with bilateral cochlear implants. While the study is well designed and executed, the clarity of the manuscript can be improved in both text and figure presentations.

Experiment design:
The description of experiment design (starting from line 164) is confusing with too much background material and omitted experiment parameters. For example, Modulation frequency for Experiment 1 is not described in the methods section. It is only mentioned in the results section (line 291). I suggest moving background to introduction or a separate paragraph in the methods and focusing on your experiment parameters in this section.

There is a great schematic of the stimuli in the supplement that is not directly referenced in the relevant methods sections. (There is a lot of material and clearer description in the supplement. Can you consider moving some material from the supplement to the main text?)

Also, was the carrier frequency set [200, 400, 800, 1600] (line 179, 403, 922, 931, 943) or [400, 800, 1200, 1600] (line 409, 437, 439, 518, figure 3A x-axis, figure 4A legend)? My guess is [400, 800, 1200, 1600] is the correct one.

Figure presentation is generally minimal and requires going back to experiment design. For example, for Figure 2 please clarify carrier frequency (4000 Hz?) and add experiment number instead of “high-frequency SAM tone”. Likewise, “400 Hz low-frequency SAM tone” can be updated to “40 Hz SAM tones with ITDfs changes (fc=400 Hz)”, as in the main text.

Adding panel titles for some figures will improve readability (for example Figure 5). Supplementary figures seem to have better panel titles (Supple Figure 5) with full description of the stimuli.

Line 40: ITDfs and ITDenv are reversed.
Line 75: There are published studies supporting the idea that ITD sensitivity is dominated by fine structure (Smith et al 2002 for example). Please add references.
Line 337: Can you spell out the main finding of Ross et al. 2007b here please?
Line 472: This section reads like a repeated beginning of the preceding section (from line 461), primarily describing Experiment 1 results.
Line 516: The amplitude of the 40-Hz ASSR gradually “decreased”?
Line 915: Figure 3A, x label should be carrier frequency (same comment for Figure 4A).
Line 924: “(A) carrier frequencies (400, 800, 1200, 1600 Hz, separated by the vertical red dotted lines)”. This line seems to have been inserted by mistake.
Figure 4: Suggest adding fc (panel A) and fm (panel B & C). Panel labels are missing in this Figure.
Supplement figure 3B. What are the open circles in this panel? They don’t appear to belong to the violin plot.

Experimental design

No comment

Validity of the findings

ACC response to modulation frequencies 40-160 Hz is overstated in the discussion (line 559), especially related to wavelet analysis. The only statement related to ACC response by ITDenv in section around Figure 5 is “The ACC responses are roughly similar in scale to neighboring brain activities, making it challenging to distinguish an ACC response even in the time-frequency domain (Figure 5B and C).” ACC response is hardly visible in Figure 2 and stated as such in the main text (line 356).

Additional comments

No comment

---

## Round 0.2 · accepted · Accept

Congratulations your manuscript has been accepted.

Reviewer 1 ·

Basic reporting

Thank you to the authors for addressing my comments carefully. I don't have any further comments.

Experimental design

I don't have any further comments.

Validity of the findings

I don't have any further comments.

Additional comments

I don't have any further comments.

Reviewer 2 ·

Basic reporting

The authors address my comments. I only noticed following minor points:

Figure 4: Please add a description of the fifth panel in the legend. Currently it is only explained in the text that the fifth panel is the overlap of the preceding panels.
Line 466, results from Figure 4?

Experimental design

no comment

Validity of the findings

no comment

Additional comments

no comment